# Management of Hematologic Malignancies in the Era of COVID-19 Pandemic: Pathogenetic Mechanisms, Impact of Obesity, Perspectives, and Challenges

**DOI:** 10.3390/cancers14102494

**Published:** 2022-05-19

**Authors:** Dimitrios Tsilingiris, Narjes Nasiri-Ansari, Nikolaos Spyrou, Faidon Magkos, Maria Dalamaga

**Affiliations:** 1First Department of Propaedeutic Internal Medicine, School of Medicine, National and Kapodistrian University of Athens, Laiko General Hospital, 17 St Thomas Street, 11527 Athens, Greece; 2Department of Biological Chemistry, School of Medicine, National and Kapodistrian University of Athens, 75 Mikras Asias, 11527 Athens, Greece; nnasiri@med.uoa.gr (N.N.-A.); madalamaga@med.uoa.gr (M.D.); 3Tisch Cancer Institute, Icahn School of Medicine at Mount Sinai, New York, NY 10029, USA; nikolaos.spyrou@mssm.edu; 4Department of Nutrition, Exercise, and Sports, University of Copenhagen, DK-2200 Frederiksberg, Denmark; fma@nexs.ku.dk

**Keywords:** blood cancer, COVID-19, hematologic malignancy, leukemia, lymphoma, multiple myeloma, myelodysplasia, obesity, SARS-CoV-2

## Abstract

**Simple Summary:**

Obesity is epidemiologically and likely, causally related to various hematological cancers, while both conditions may predispose to severe SARS-CoV-2 infection. The COVID-19 pandemic brought about a variety of obstacles with respect to numerous aspects in the management of hematological malignancies. In the present overview, the evidence linking obesity with the development of hematological cancers and their role as risk factors for severe COVID-19 is critically appraised. Furthermore, the various challenges which emerged during the the pandemic are reviewed, regarding not only the treatment of the underlying hematological malignancies themselves, but also the prevention and therapeutic management of SARS-CoV-2 infection in this patient group. Lastly, we discuss further unresolved issues which need to be addressed in order to optimize the management of patients with hematologic cancers in the course of ongoing COVID-19 pandemic.

**Abstract:**

The COVID-19 pandemic brought about an unprecedented societal and healthcare system crisis, considerably affecting healthcare workers and patients, particularly those with chronic diseases. Patients with hematologic malignancies faced a variety of challenges, pertinent to the nature of an underlying hematologic disorder itself as well as its therapy as a risk factor for severe SARS-CoV-2 infection, suboptimal vaccine efficacy and the need for uninterrupted medical observation and continued therapy. Obesity constitutes another factor which was acknowledged since the early days of the pandemic that predisposed people to severe COVID-19, and shares a likely causal link with the pathogenesis of a broad spectrum of hematologic cancers. We review here the epidemiologic and pathogenetic features that obesity and hematologic malignancies share, as well as potential mutual pathophysiological links predisposing people to a more severe SARS-CoV-2 course. Additionally, we attempt to present the existing evidence on the multi-faceted crucial challenges that had to be overcome in this diverse patient group and discuss further unresolved questions and future challenges for the management of hematologic malignancies in the era of COVID-19.

## 1. Introduction

The outbreak of the COVID-19 pandemic in December 2019 brought about an unprecedented global crisis at health care, societal and economical levels. The subsequent allocation of personnel- and material-related resources towards countering COVID-19 endangered the uninterrupted care of patients with a large variety of chronic health conditions, potentially compromising the overall prognosis, particularly of those in danger of severe SARS-CoV-2 infection [1,2].

Since early in the course of the pandemic, the presence of obesity or hematologic malignancy, among others, were identified as important risk factors for SARS-CoV-2 acquisition and severe infection [3,4,5,6]. Interestingly, an epidemiologic and probably pathogenetic connection between these two entities has long been established [7,8,9].

The inherent pathophysiological features of a variety of hematologic neoplasms, the associated changes in the number and/or function of various immune cells and occasionally, the effects of therapy, all have a profound impact on underlying immune mechanisms. As a result, affected individuals form a patient group particularly susceptible to infectious complications including COVID-19. Additionally, although highly variable depending on hematologic diagnosis, the initial and long-term management of several hematologic cancers exhibits numerous challenges regarding many aspects and demands a meticulous adherence to a strict treatment regimen and follow-up schedule, as well as a collaboration of different medical disciplines at various levels of the health care system. Not surprisingly, the combination of these features has made the management of patients with hematologic malignancies a particularly demanding task during the course of the pandemic.

In the present narrative review, the proposed mechanisms linking obesity with hematologic carcinogenesis and the spectrum of related diagnoses are discussed. Furthermore, the obstacles and challenges that emerged during COVID-19 regarding the management of these patients, and the corresponding measures to meet them, are discussed along with a review of open questions and aspects necessitating optimization.

## 2. Overview of Obesity-Related Hematologic Neoplasms

A relationship between increased body adiposity and elevated risk of a variety of hematologic neoplasms has long been established, expanding the knowledge linking obesity to the development of solid neoplasms [7,8]. Relevant evidence has derived primarily from a considerable number of population-based cohorts subjected to long-term epidemiologic observation (Table 1).

The risk extends to an abundance of difference hematologic malignancies, including non-Hodgkin lymphomas (NHL) [10,18,20], Hodgkin’s disease [10,11], multiple myeloma [18,19], and acute leukemia [20] lymphocytic (ALL) and myeloid (AML) [12,13] (Table 1). The impact of obesity on the risk of NHL may not be homogenous across different histological subtypes, and the increased risk appears to concern primarily the more aggressive diffuse large B-cell lymphoma and, to a much lesser extent, the more indolent subtypes, such as follicular lymphoma or small lymphocytic lymphoma/chronic lymphocytic leukemia [27,28].

Furthermore, there is an increased prevalence of pre-malignant hematologic conditions among individuals with obesity, such as the spectrum of myelodysplastic syndromes (MDS), for which obesity is also associated with a worse prognosis regarding overall survival [21,29]; chronic myeloproliferative disorders, especially chronic myeloid leukemia and essential thrombocytosis [23,24,25,26]; and monoclonal gammopathy of unknown significance (MGUS) [14]. The probability of progression of monoclonal gammopathy of unknown significance to multiple myeloma also appears to be strongly affected by the presence of overweight and obesity [14,30].

The fact that the bulk of available data has been generated in fairly well-characterized cohorts, which enables their extraction to be independent of potential confounding factors, does not by itself conclusively prove a causal role of obesity and related systemic perturbations in hematologic oncogenesis. On the contrary, it could be argued that these observations may be at least partly attributable to unaccounted dietary or other lifestyle factors, or other genetic and environmental factors that promote both conditions. The major counterargument against this hypothesis stems from the decrease in total and histologic type-specific malignancy risks following conservative or surgical weight loss [31,32,33]. In the frame of the current discussion, it should be noted that evidence on an incidence reduction specific of hematologic (pre-)malignancies is, to date, extremely scarce, although a reduction in NHL occurrence among females with obesity undergoing bariatric surgery has been reported [33].

Not surprisingly, in most available clinical studies that have verified the association between obesity and hematologic (pre)malignancies, the body mass index (BMI) has been used to characterize and quantify the degree of adiposity among participants. Both the universally accepted cutoffs for overweight (25 kg/m^2^) and obesity (30 kg/m^2^) and the BMI itself as a continuous parameter have been used to quantify the relationship between the severity of obesity and malignancy risk. Despite its simplicity, BMI is an imperfect tool for the assessment of the qualitative features of adiposity. BMI values do not distinguish between lean and fat mass, while they do not take into consideration differences in fat distribution (including the fractions of visceral and subcutaneous adipose tissue) and the age-related decline of lean mass [34,35]. Although data on other body composition markers are scarce, indices of visceral adiposity, such as waist circumference and body-shape-index as well as waist-to-hip ratio, have been shown be better predictors of the development of any hematologic malignancy and multiple myeloma, respectively [36]. The stronger implication of clinical markers typically associated with an “unhealthier” obese phenotype may imply that the already well-characterized adverse metabolic and overall consequences of increased total obesity and particularly central obesity might be causally implicated in hematologic carcinogenesis [36].

On the other hand, the obesity-related hematologic malignancies and premalignant conditions constitute an extreme diverse collective with respect to their cell line of origin, other risk factors for their development, clinical presentation, prognosis and therapy. Consequently, a unified hypothesis linking their pathogenesis to obesity would be challenging to establish. Among the mechanisms that have been proposed to link obesity with carcinogenesis, the most predominant include aberrant cytokine and adipokine production from the expanded and dysfunctional adipose tissue, increased circulating levels of peptide hormones with growth-factor properties (e.g., insulin, IGF-1) and the development of a systemic oxidative-stress environment promoting DNA-damage and tumor genesis [8,37,38,39,40,41,42,43,44,45]. The chronic immune stimulation in the frame of the low-grade inflammatory milieu of the obese states may be of particular significance in the case of hematologic neoplasms. Adipose tissue-derived proinflammatory cytokines (e.g., intereukin-6, TNF-alpha) have prominent direct effects on hematopoietic cell proliferation, evasion of apoptosis, kinetics and function [46,47,48,49,50]. Furthermore, altered secretion of members of the adipokine family, most predominantly increased leptin secretion and function and decreased adiponectin, may play a distinct role in obesity-related hematologic carcinogenesis. Leptin partakes in the normal hematopoietic function [51], while exerting antiapoptotic and/or proliferating effects on lymphocytes [52] and myeloblasts through binding to its receptors on the aforementioned cell types [53]. On the other hand, lower adiponectin levels are independently associated with a higher risk of myelodysplastic syndrome, acute myelogenous leukemia and multiple myeloma, among others [8,41,42,44,54]. It is likely that not all obesity-related hematologic malignancies are influenced to the same degree by specific mechanisms; rather, potential combined effects of these factors may be decisive for the oncogenesis of specific hematologic neoplasms.

## 3. Association between Obesity and Hematologic Cancers with COVID-19 Outcomes

### 3.1. COVID-19 and Obesity

Since the beginning of the pandemic, obesity was recognized as a major factor associated with SARS-CoV-2 infection adverse outcomes [55,56]. From a meta-analysis of 30 early studies that reported specific outcomes with respect to measures of adiposity, an overall adjusted odds ratio (OR) of 2.09 (1.67, 2.62) for severe COVID-19 among patients with a “high” compared with those with a low BMI was noted. The adjusted ORs for individual outcomes were 2.36 for hospitalization, 2.32 for intensive care unit (ICU) admission, 2.63 for mechanical ventilation and 1.49 for death [5]. Additionally, a considerably higher visceral adipose tissue (VAT) accumulation was ascertained among those experiencing adverse outcomes compared with those with mild disease by three studies [57,58,59]. It should be noted that even though most included studies used the conventional cut off of 30 kg/m^2^, there were others that deviated from this criterion to discriminate between patients with high and low BMI, whereas no clear definition was given in six studies [5]. Later studies confirmed these findings, further highlighting a dose-dependent relationship between the severity of obesity and risk for most adverse COVID-19 outcomes [60,61,62]. Interestingly, the overall relationship of BMI with serious SARS-CoV-2 infection conforms to a J-shaped curve, with those at the lowest adiposity extreme being more prone to an adverse course, hence, highlighting another manifestation of the “Obesity paradox” in the frame of COVID-19 [60,61].

### 3.2. COVID-19 Outcomes in Obesity-Related Hematologic Malignancies

An overview of the studies on the impact of hematologic malignancies on SARS-CoV-2 outcomes is presented in Table 2.

The earliest evidence regarding SARS-CoV-2 infection in patients with hematologic malignancies originated from a small cohort study among hospitalized patients in hematologic wards at the time of the initial outbreak in Wuhan, China (Table 2). There were similar SARS-CoV-2 acquisition rates among patients compared with a matched controlled group of health care workers, but a higher incidence of severe disease and death [4].

Furthermore, an analysis of the data of the openSAFELY platform in the United Kingdom pointed towards increased mortality among patients with hematologic malignancies. The risk was highest (2.5-fold) among those who had received the diagnosis in the last 1–5 years and slightly decreased thereafter, either reflecting the acute detrimental effects of treatment or resulting from survivalist bias among patients with greater survival rates [70]. A subsequent meta-analysis of 39 reports (5 pediatric and 34 among adults) attempted a more comprehensive evaluation of the effects of age, treatment and specific diagnosis on the risk of death from COVID-19. The risk of death in adults was found to be higher than in pediatric hematologic patients (34% vs. 4%), and among those, higher for those aged >60 (relative risk: RR 1.8) as well as lower among white compared to non-white patients (RR 2.2). The pooled risk was substantially higher than this observed in the general population. The decisive effect of age on COVID-19 outcomes observed in this study is in agreement with the vast majority of available reports in diverse populations. The relationship between age and the risk of hospitalization or death due to SARS-CoV-2 infection appears to be linear, without distinct age cut-offs, above which an additional risk for severe disease is conferred [71]. Furthermore, it is likely that the effect of age is not mediated by the increased co-morbidity burden in the elderly population [72]. Although the factors that drive this relationship are not fully elucidated, a crucial role of the age-related immune senescence may be hypothesized [73]. Likewise, an increased risk for severe disease among black patients and racial minorities compared to white patients has been also observed in numerous cohorts. There are likely to be numerous and diverse factors that could explain the racial disparities regarding COVID-19 outcomes. These include a compromised socioeconomic status and poorer living/working conditions, limited or delayed access to health care services, higher prevalence of relevant comorbidities, poor nutritional habits, more frequent smoking and increased psychosocial stress [74,75]. Interestingly, the presence of any or solely cytotoxic recent anticancer therapy was not shown to have an effect on mortality compared to the absence of treatment. It is also noteworthy that, excluding patients with acquired bone marrow failure syndrome (commonly in the frame of MDS or aplastic anemia) which showed a peak mortality rate of 53%, the risk of death was relatively homogenous for the rest of the diagnosis, ranging from 31–33% for those with CLL, lymphomas and MM to 41% for those with acute leukemias [67].

In a more recent report from a nationwide database from the USA, SARS-CoV-2 acquisition, hospitalization and death rates were investigated across a broad spectrum of malignant and premalignant hematologic conditions and were compared to patients without haemato-oncologic diagnoses [68]. Notably, the sum of investigated conditions concerned predominantly obesity-related hematologic malignancies (Table 1). Acquisition rates were substantially higher among patients within a broad spectrum of diagnoses, especially among those who had received a hematologic diagnosis within the last year (pooled adjusted OR 11.9 [11.3–12.5]). Maximal rates were observed in patients with ALL (aOR 31.0) but they were also substantially high across the sum of studied conditions. It cannot be definitely concluded whether, and to which degree, this finding reflects a truly increased risk of viral acquisition or a greater probability of developing symptoms, hence, more frequently diagnosed SARS-CoV-2 infections. Likewise, hospitalization rates and mortality among recently diagnosed hematologic patients with COVID-19 were higher than those without a diagnosis and those with a recent hematologic diagnosis without COVID-19 (51.9% vs. 23.5% and 15% and 14.8% vs. 5.1% and 4.1%, respectively). Findings were comparable when accounting for any time of hematologic diagnosis. Similar to previous reports, patients of African-American origin exhibited generally higher rates of adverse outcomes in comparison with Caucasians.

## 4. Obesity, Related Hematologic Malignancies and Severe COVID-19: Pathogenetic Considerations

It may not come as a surprise that two epidemiologically- and possibly causally-related conditions—obesity and hematologic malignancies—both belong to the known factors predisposing individuals to adverse COVID-19 prognoses (Figure 1). The organ-specific, obesity-related comorbidities—such as diabetes mellitus and arterial hypertension—that established cardiovascular and restrictive pulmonary disease on one hand, and the overall frailty that characterizes certain groups of individuals affected with hematologic cancers on the other, predispose people to adverse outcomes in acute illnesses of various causes, including sepsis [76,77,78,79,80]. Nonetheless, specific features of the systemic immune response in the frame of SARS-CoV-2 infection can be hypothesized to influence the course of the disease among individuals affected with obesity and/or hematologic malignancies.

Successful host defense and viral clearance requires a multifaceted innate and acquired cellular immune response, orchestrated by a complex variety of cellular and humoral components. Of crucial importance are a timely pro- and anti-inflammatory cytokine cascade and the balanced involvement of CD4+/CD8+ and B lymphocytes. An aberrantly exaggerated, uncoordinated immune response may drive the elusive manifestation of the hyperinflammation syndrome [81].

Quantitative and qualitative defects of the immune system are most prominent in hematologic malignancies but are encountered in obesity as well [82]. Lymphopenia is a prominent feature of a broad variety of hematologic malignancies such as MDS [83], Hodgkin disease, NHL [84] and MM [85], and can complicate therapy. ALL can be viewed as a state of “functional” lymphopenia despite the occasionally extremely high lymphocyte counts. Quantitative and qualitative T-lymphocyte defects have also been described in obesity and may be partially reversible following successful weight loss [86]. Interestingly, the association between lymphopenia and adverse SARS-CoV-2 outcomes, including hospitalization, ARDS, mechanical ventilation, and death, has been observed in numerous studies [87,88,89]. Besides being a consequence of SARS-CoV-2 infection itself [90], which may imply a component of reverse causality for this observation, pre-existing lymphopenia has also been shown to predispose people to severe COVID-19 [91], which advocates for a crucial role of T cell-mediated immune response for successful viral clearance and infection resolution. Furthermore, lymphopenia, either preceding or developing in the course of SARS-CoV-2 infection, may facilitate the development and/or increase the severity of secondary bacterial and/or fungal infections that may further adversely affect prognosis [92,93]. Alternatively, a prominent reduction in regulatory T cells could predispose people to an exaggerated immune response and cytokine storm which are major features of the COVID-19-related hyperinflammation syndrome [94]. A state of systemic chronic low-grade inflammation is a feature of both obesity and various hematologic malignant states, such as MDS, acute leukemias, MM and NHL [95,96,97,98]. This feature can promote or be inherent in “inflammaging”, which refers to a state of immune cell senescence and dysregulation [99], predisposing to adverse COVID-19-related outcomes [99,100]. Lymphocyte telomere shortening constitutes an important component of lymphocyte senescence and dysfunction, which is typically encountered in hematologic malignancies, occasionally in conjunction with an adverse prognosis [101]. Telomere length has been also shown to inversely associate with BMI [102] and correlates with the negative metabolic consequences of excess adiposity [103], while weight loss causes a dose-dependent elongation [99]. A shorter telomere length has been suggested as the epidemiologic link between known clinical and epidemiologic risk factors for severe COVID-19, implicating impaired lymphocyte replication and recruitment capacity as well as the senescence-associated pro-inflammatory lymphocyte phenotype in severe disease pathogenesis [104,105]. Indeed, a shorter telomere length has been associated with poor SARS-CoV-2 infection outcomes [106,107], while the infection itself may have an accelerating impact on telomere shortening [108].

The release of extracellular traps (NETs) by neutrophils is increased in certain hematologic cancers such as DLBCL [109] or CLL [110] as well as in severe obesity [111]. NETosis has been shown to be a prominent feature of COVID-19 immunothrombosis, which is implicated in target organ damage [112]. Likewise, aberrant complement activation constitutes another common feature of the three entities [112,113,114].

## 5. Vaccination against SARS-CoV-2 in Patients with Hematologic Malignancy

Upon the market release of COVID-19 vaccines in December 2020 and in the face of initial relative shortages, there was an urgent need for a rationalized prioritization of vaccine supplies. The aim was to maximize the efficacy of vaccination strategies with respect to maximal prevention of COVID-19-related deaths, reduction in hospitalization burden and societal and economic disruption. The World Health Organization issued corresponding guidance to direct available vaccine resources to the most vulnerable patient groups [115]. Patients with comorbidities rendering them vulnerable to severe disease were categorized in the second priority group; hence, these patients were scheduled to be vaccinated in the first 11–20% of the population. Although individual national programs occasionally deviated from this scheme, both obesity as well as hematologic malignancy fell into this category, the latter corresponding to “cancer” as well as to “conditions or therapies associated with immune suppression” in the issued document [115]. Similar approaches were used for the prioritization of booster vaccinations and are likely to be repeated in the future updated versions of vaccination programs as the pandemic continues.

The vaccines critically altered the course of the pandemic, being highly efficacious against symptomatic SARS-CoV-2 infection and, even after the appearance of altered viral variants, for the prevention of severe course, hospitalization and death. Despite initial considerations, vaccine efficacy was shown to be essentially unaffected by the presence of obesity [116,117]. Vaccine efficacy among those with obesity-related hematologic cancers is inherently more challenging since it is expected that the disturbed immune function originating from the conditions themselves or their therapies may hinder the development of adequate immunity. Indeed, inadequate antibody and/or T-cellular responses have been documented after the vaccination of patients with monoclonal gammopathies (particularly for MM rather than smoldering myeloma [118]), acute leukemias, Hodgkin disease and NHL, MDS, acute leukemias and chronic myeloproliferative disorders [118,119,120]. Seroconversion rates appear substantially lower among hematologic cancer patients compared with those affected by solid neoplasms [120,121], and may be particularly low in specific diseases, depending on their pathophysiological and cytogenetic characteristics as well as mode of therapy. Greenberger et al. investigated the antibody responses following two mRNA-based platform doses in 1445 hematologic patients from the USA, approximately 40 days following the second dose. Seropositivity rates were found to be minimal across a variety of NHL histological subtypes or CLL and, in contrast, were satisfactory in Hodgkin disease, acute leukemias, monoclonal gammopathies and myeloproliferative disorders [122]. Unsurprisingly, B-cell-targeted therapies such as anti CD20-monoclonal antibodies, chimeric antigen receptor (CAR) T cells against CD19, or Bruton’s tyrosine kinase (BTK) inhibitors have been associated with diminished antibody responses [120,121,122]. Regarding receivers of allogeneic hematopoietic stem cell transplantation (HSCT) and despite some concerns [121], in a cohort of 117 patients vaccinated with BNT162b2, antibody response rates were satisfactory among patients who had received the transplantation more than one or two years ago, compared with less than 12 months. Response rates after two doses were 89%, 96%, 52%, respectively, and were greater among those with lymphocyte counts of >1000/μL compared with those with a lower lymphocyte count (91% vs. 64%, respectively) and those who were not under ongoing treatment (91% vs. 64%, respectively) [123]. Likewise, a study among 67 lymphoma or MM patients who received autologous HSCT showed that 56 (87%) developed humoral [124] immunity following the vaccination of the BNT162b2 mRNA vaccine. Similar findings have been ascertained in a case series of scleroderma patients who received autologous stem cell transplantation [125]. Apart from diagnosis and mode of therapy, other factors acknowledged to positively affect the immune response include vaccination outside the frame of active therapy, a longer time interval since the last anti-CD20 antibody administration, and vaccination during a complete remission status [120,126].

Regarding vaccine type, there have been occasional reports of favorable efficacy profiles of specific platforms; between the two available mRNA-based preparations, higher antibody titers may be observed among patients with monoclonal gammopathies after receiving the mRNA-1273 than the BNT162b2 vaccine [118]. Likewise, in the study by Greenberger et al., the vaccination with mRNA-1273 was associated with greater odds of seropositivity status compared with BNT162b2, independent of multiple confounders, including the type of hematologic malignancy [122]. Although there is a limited number of studies to address specific vaccines among hematologic patients and available reports vary with respect to participant composition and studied circumstances, there were no significant differences in seropositivity rates across the different available vaccine platforms according to a recent meta-analysis [120]. Subsequently, although it does not seem justified based on the current data to prioritize one vaccine platform over another for its use among hematologic patients, more information in hematologic populations according to specific underlying diagnosis and treatment is needed to optimize vaccination strategies. In any case, adherence to a timely vaccination schedule irrespective of a specific platform for all affected patients is strongly warranted.

It should be noted that most available data regarding vaccine immunogenicity have been based on induced neutralizing antibody titers. Undoubtfully, even among individuals who show suboptimal or even negative antibody levels following vaccination or SARS-CoV-2 infection, a certain cellular immunity-mediated protection against symptomatic infection and severe disease may be assumed. However, based on available evidence, it seems likely that neutralizing antibody responses themselves correlate satisfactorily with the probability of SARS-CoV-2 infection [127,128].

At least a third (booster) vaccine dose is recommended in most countries for individuals who have successfully undergone a full vaccination course to overcome the waning of immunity and prevent breakthrough infections, especially by new variants. Immunocompromised individuals are to be prioritized in this strategy, while local guidelines occasionally advise in favor of a second booster (fourth) dose for these patients [129], despite the extremely scarce currently available evidence to support the effectiveness of this recommendation [130,131]. Patients with hematologic malignancy would fall into this category of high-risk patients. Available data on booster dosing demonstrate that despite the fact that a third dose of BNT162b2 strengthens humoral immunity among antibody-positive hematologic patients, it does not alter the serological status among those that remained seronegative despite prior immunization; nevertheless, it does seem to augment T cell-mediated cellular immunity irrespective of antibody status [132], thus, demonstrating a potential benefit of booster immunization even among those with a prior insufficient humoral response.

## 6. Blood Product Transfusion in the Era of COVID-19

Individuals with hematologic malignancies constitute a pool of patients with urgent and frequent dependency on the transfusion of blood products. Voluminal requirements greatly vary depending on underlying diagnosis, type of treatment and local standard operating procedures of hematology oncology units; nonetheless, transfusion of erythrocytes and/or platelets is frequently necessary in the setting of active cytotoxic therapy, while in the case of certain background diagnoses (e.g., particularly low-risk MDS), erythrocyte and/or platelet transfusions constitute the core of long-term maintenance therapy [133].

Concerns regarding the potential infectivity of blood units from asymptomatic or pre-symptomatic infected donors originate from the ascertainment of viral RNAemia in hospitalized patients in some reports, although this finding is not universal among available studies [134]. Nevertheless, in a broad screening of plasma minipools from 258,000 blood donations between March and September 2020 in the US, positive SARS-CoV-2 PCRs were remarkably uncommon and generally in very low titers [134]. Real-world data from cases of blood product transfusion donated by pre-symptomatic infected patients also support the notion that even though transfusion as a mode of SARS-CoV-2 transmission may be theoretically plausible, it is extremely rare [135,136,137,138,139].

Particularly during the initial outbreak of the pandemic, but also during every subsequent accelerated phase leading to a new surge in infections, there was a visible danger of blood supply shortages. Shortages may develop depending on geographical region, hospital level of care and resource availability as a result of reduced donation rates due to a fear of donors being in contact with healthcare services, attending transfusions, physician concerns in the face of the presence of COVID-19-suspicious donor symptoms, increased demand for blood products to cover the needs of hospitalized COVID-19 patients, or an overall reassignment of personnel- and material-related resources to counter the pandemic waves. In the case of a development of extreme shortages, it is likely that updated guidelines by competent authorities would be necessary regarding a re-evaluation of transfusion intervals and/or thresholds [140,141] with the aim of rationalizing the use of available resources.

Despite initial concerns [141,142], and even though the collective volume of blood donations and success of donor recruitment campaigns reduced during the first pandemic waves [143,144,145,146], this was counterbalanced by a reduction in blood product demand, chiefly through a reduction in elective surgical procedures and medical transfusions [147]. Although some blood product supply shortage did develop in some regions [148,149], there is no available evidence that this phenomenon had a significant impact on planned or acute transfusions among patients with hematologic disease [143].

Nevertheless, the degree of shortages and resulting initial uncertainties brought about by COVID-19 have provided further reasons for the reappraisal and optimization of current transfusion strategies. This would concern not only the regular transfusion programs among hematologic patients, but also the transfusions for practically any medical indication in a variety of, at least non-acute, circumstances. With respect to chronic hematologic diseases, caution should be taken in the detection of reversible causes of anemia, such as iron, folic acid, or vitamin B12 deficiency, as well as their treatment through adequate oral or parenteral substitution. Furthermore, the indications for erythropoietin therapy should be meticulously followed or even broadened in the face of emerging new evidence in the long term, since its administration may reduce volume requirements in transfusion-dependent patients [150]. Importantly, lower transfusion thresholds than commonly utilized seem to be safe and well-tolerated among asymptomatic patients [151], thus, restrictive rather than liberal transfusion strategies may be preferred in order to optimally portion erythrocyte supplies. A similar strategy could also apply for stable, non-bleeding patients with chronic thrombocytopenia dependent on thrombocyte transfusions. Furthermore, accessory measures could apply to transfusions for surgical indications, such as the correction of disorders of hemostasis prior to elective surgical procedures, intra- or postoperative use of coagulation factor concentrates, antifibrinolytic agents such as tranexamic acid and utilization of autologous blood transfusion where indicated. Lastly, frequency and volume of blood draws in blood supply storage should be limited to the absolute minimum that may affect clinical decision making, both in the acute and chronic setting.

## 7. Medical Therapy against COVID-19 among Individuals with Hematologic Malignancies

Although broadly applied vaccination programs have been and continue to constitute the mainstay of strategies to counter the COVID-19 pandemic, there has been a continuing worldwide effort for the development of drug therapies against the SARS-CoV-2 infection, either through the development of new agents or through repurposing already available drugs marketed for other indications [152,153]. Certain national recommendations for the treatment of the SARS-CoV-2 infection according to its severity take into account the comorbidity background of affected patients to guide and possibly escalate medical management [154,155]. Accordingly, patients with hematologic malignancies, particularly those receiving specific therapies, among others, are to be prioritized under the conditions of logistical or supply constraints which may preclude the universal use of anti-SARS-CoV-2 therapies; these patients are at an increased risk of severe disease, as well as they are not able to mount an inadequate immune response after vaccination [154].

The already available and constantly evolving body of relevant evidence contributes to the World Health Organization’s living guideline on drugs for COVID-19 [156]. In this guideline, recommendations for medical therapy are formed based on evidence from clinical studies and are escalated according to disease severity (characterized based on clinical criteria as mild, severe or critical). Although there is no specific reference to hematologic cancers or any other specific patient group for that matter, certain recommendations are shaped based on a perceived underlying higher risk for hospitalization based on coexisting medical conditions, a case that arguably also concerns patients with HMs. According to the living guideline of the World Health Organization, as of May 2022, viral protease inhibitors Nirmatrelvir/Ritonavir and conditionally Molnupiravir, monoclonal antibodies such as Sotrovimab and Casirivimab/Imdevimab, and Remdesivir are recommended therapies for patients with non-severe disease; for those with severe or critical COVID-19, corticosteroids, Janus kinase inhibitor Baricitinib, interleukin-6 inhibitors and conditionally Casirivimab/Imdevimab are indicated [156]. In contrast, the use of ivermectin or convalescent plasma outside the context of clinical trials are now discouraged [156].

In the absence of specific recommendations for this population, COVID-19 therapy in hematologic cancer patients is to be guided based on available general recommendations. However, there are points that pertain to the unique characteristics of these patients that may ought to be taken into account, including current and cumulative glucocorticoid doses, presence of hypogammaglobulinemia, perceived degree of immune suppression due to the illness or its therapy, cytopenic status due to the underlying condition or the risk of cytopenia from certain COVID-19 medical treatments (e.g., Remdesivir, Molnupiravir). Various therapies or combinations have been successfully used among hematologic patients [157,158,159,160]. In a series of 313 patients with hematologic malignancies from Israel, Remdesivir administration was associated with decreased mortality [65]. Furthermore, in a subgroup of 216 patients with very severe disease from an initial sample of 367 pediatric and adult patients with hematologic malignancies, azithromycin and corticosteroid administration were associated with lower 45-day mortality (OR 0.42 (95% CI 0.2–0.89) and 0.31 (0.11–0.87), respectively). In addition, three cohort studies have demonstrated potential benefits including duration of hospitalization, course severity and survival after administration of convalescent plasma in patients with hematologic malignancies [161,162,163]. In the largest of those, Thomson et al. investigated a sample of 966 hospitalized hemato-oncologic patients, 143 of whom received convalescent plasma therapy. Even after adjustment for confounders or propensity score matching, 30-day mortality was lower among receivers (HRs 0.60 (0.37–0.97) and 0.52 (0.29–0.92), respectively), while the effects were even more pronounced among patients admitted to the ICU or receiving mechanical ventilation (HRs 0.40 (0.20–0.80) and 0.32 (0.14–0.72), respectively) [163]. These findings contradict the neutral results of convalescent plasma therapy in other populations, most strikingly in the Randomized Evaluation of COVID-19 Therapy (RECOVERY) trial [164]. Although limited by their non-randomized design, cohort studies should be taken into account, and it could be postulated that this discrepancy may reflect the benefit of exogenous immunoglobulin administration in the setting of hypogammaglobulinemia, which often occurs as a result of certain hematologic malignancies such as CLL or MM or their therapies [165].

Despite scarce evidence regarding potential benefits from specific therapies in hemato-oncologic patients, most available recommendations do not offer specific guidance for the treatment of the SARS-CoV-2 infection in this patient group. Nevertheless, given the strongly immune-suppressing nature of many of the conditions themselves or their treatments, additional caution is warranted regarding the use of therapies against COVID-19 with further immune-modulating modes of action (i.e., corticosteroids, intereukin-6 antagonists, interleukin-1 antagonists-anakinra [166]).

## 8. Influence of the COVID-19 Pandemic on the Treatment of Malignant Hematologic Disease

The outbreak of COVID-19 introduced significant challenges for the care of patients with actively managed hematologic malignancies. Particularly during the initial stages of the pandemic, uncertainties with respect to factors that predispose people to SARS-CoV-2 acquisition and severe course, as well as the lack of effective treatments and vaccines, were added to concerns arising from the frail immune and general poor clinical condition of a large proportion of acutely treated hematologic patients. These may confer increased susceptibility to respiratory viral infections, whereas lymphopenia, as well as specific modes of therapy such as purine analogs anti-CD20 or CD52 and allogeneic HSCT, may pose a particularly increased risk [167]. Additionally, a common adverse event of CAR T-cell therapy is the cytokine release syndrome, attributed to the abrupt secretion of inflammatory mediators by CAR T cells and other cell types [168]. The presence of a similar physiological state of cytokine hypersecretion has been reported to predispose people to a severe SARS-CoV-2 infection course [169]. Interestingly, therapy with tocilizumab, which is an integral component of COVID-19 therapeutic regimens, is also used to treat CAR T cell-related cytokine release syndrome [170].

These concerns, among others, led to initial considerations of delaying non-urgent hematologic malignancy treatment, including autologous HSCT among standard-risk individuals, in order to avoid exposing patients not only to the risks of therapy, but also to reduce the frequency of non-urgent contacts with the healthcare system for treatment and monitoring blood draws [171]. A similar approach has been generally adopted and generalized by numerous national and international hematologic societies. Guidelines vary according to issuing authorities and on the basis of hematologic diagnosis.

A detailed overview of the sum of specific treatment recommendations in the frame of COVID-19 is beyond the scope of the current review. Testing for SARS-CoV-2 and, occasionally, chest computed tomography, is indicated for patients before the initiation of each therapy cycle, in donors and recipients before HSCT as well as in the presence of COVID-19-suspicious clinical symptoms [172,173,174,175]. A general pattern of treatment initiation or continuation without delay exists for patients who are considered high risk/high priority within or among specific diagnoses. This also includes allogenic HSCT when carried out with curative intent in an urgent indication (e.g., high-risk ALL or AML [173,175]). Autologous SCT may be delayed for no longer than 8 weeks in less urgent clinical settings (e.g., in eligible MM patients [175,176]). Where possible, highly immunosuppressive alternatives (e.g., schemes including bendamustine) should be avoided [173], and where possible, the lowest effective doses of agents, which may induce aplasia (e.g., methotrexate, cytarabine), should be administered [173]. Appropriate schemes that can be administered in outpatient settings may be preferred [173,175].

Preventive measures of SARS-CoV-2 acquisition should be strictly adhered to [172]. An occurrence of SARS-CoV-2 infection amidst active treatment generally mandates a pause of systemic anticancer therapy and/or delay of allogeneic HSCT until laboratory resolution (negative PCR test) [173,175], although there are isolated reports of continued modified systemic treatment despite SARS-CoV-2 infection, albeit of noncritical severity [177]. As is the case with any deviation from hematologic cancer treatment protocols, this decision should be weighed against the risk of therapy delay on a case-by-case basis [176]. Non-systemic antineoplastic administration (such as intrathecal chemotherapy) or non-cytotoxic, non-immunosuppressive systemic therapies such as tyrosine kinase inhibitors may be continued [173]. Evidently, the decision to administer any kind of anticancer treatment during the course of active COVID-19 warrants additional caution for the avoidance of interactions with anti-SARS-CoV-2 therapies and subsequent drug-related adverse events [173].

Despite general recommendations and development of guidelines that cover a broad range of clinical scenarios, the field of hematologic cancer therapy during the pandemic unavoidably includes a highly individualized component regarding clinical decision-making. A great degree of clinical readiness and experience is needed from the side of involved clinicians, in conjunction with adherence to local standard operating procedures and firm cooperation with hygiene and infectious disease specialists.

## 9. Lifestyle Changes (Diet and Exercise) and Cancer Survivorship in the Era of the COVID-19 Pandemic

Several studies have demonstrated a shift towards healthier behavior after cancer diagnosis or treatment [178,179,180,181,182,183]. Whereas the majority of evidence on lifestyle changes after cancer diagnosis or treatment has been drawn from female breast cancer survivor cohorts [180,183,184], several studies have focused on hematologic malignancies [180,185,186,187,188]. A recent cross-sectional study demonstrated that hematologic malignancy survivors exhibit a significant reduction in smoking behavior, with many patients quitting and nearly all patients limiting their smoking behavior after cancer diagnosis [180]. The researchers were not able to detect a significant change in exercise or dietary habits after diagnosis [180]. High post-diagnosis physical activity was associated with lower mortality among hematologic cancer survivors in a large retrospective analysis of 5182 patients [187]. Specifically, in a multivariate analysis including all hematologic cancer survivors, physical activity after diagnosis yielded an all-cause mortality hazard ratio of 0.61 (95% CI: 0.50–0.74), and the association remained significant in the subgroups of NHL, myeloma and leukemia survivors, respectively. A secondary analysis of a randomized controlled trial in allogeneic stem cell transplant patients compared the survival outcomes between patients assigned to an exercise intervention or a control group [180]. Total mortality (TM) but not non-relapse mortality (NRM) risk was reduced among the 103 allogeneic stem cell transplant patients with a combination of endurance and resistance exercise (TM: exercise vs. control group: 12.0 vs. 28.3%, *p* = 0.03) over a 2-year follow-up period. In a randomized controlled setting, the Healthy Exercise for Lymphoma Patients (HELP) trial randomized 122 lymphoma patients to either supervised aerobic training or no exercise [185]. The trial analysis did not detect a positive effect after 12 weeks of exercise on the progression-free survival of 122 lymphoma patients during a 61-month follow-up period. In a recent clinical trial, the researchers hypothesized that caloric restriction and increased exercise would not only decrease fat mass but would also lead to reduced post-induction minimal residual disease in patients with B-cell acute lymphoblastic leukemia [186]. The intervention consisted of caloric deficits of ≥20% during induction, reduced fat intake, reduced glycemic load, and increased exercise; this led to a reduction in fat gain in the overweight and obese patients, and most importantly, it reduced minimal residual disease. Nevertheless, this single-arm intervention study utilized historical controls for comparative purposes and prospective randomized trials are eagerly needed for the validation of these findings.

In conclusion, several studies suggest that high physical activity levels are of potential benefit to the survivors of hematologic malignancies. However, physical activity levels can be affected by the overall condition of the patient, and the intensity of the treatment regimen or treatment-related complications; these factors can also directly affect survival outcomes. Larger, well-controlled clinical trials are required to unravel the potential causal relationship between behavioral changes and long-term outcomes in hematologic malignancy survivors.

The recently worldwide-imposed lockdowns during the COVID-19 pandemic had a dramatic impact on the lifestyle of the general population, including cancer survivors. An exploratory study in Northern Ireland reported the effects of the pandemic on several behavioral aspects of cancer survivors during the COVID-19 pandemic [189]. Interestingly, 52% of the patients reported reduced physical activity during restrictions, owing to isolation, declining health, lack of access to exercise facilities and motivation to exercise. Whereas 4% of participants had reported no physical activity before the pandemic, this number increased to 21% during the COVID-19 restrictions. In another observational study in France, 64% of the cancer survivors reported reduced physical activity and increased screen time during the COVID-19 lockdowns [190]. On the other hand, researchers from Italy effectively utilized tracker-based counselling and live-web exercise to engage breast cancer survivors in exercise and improve their sleep quality during the COVID-19 lockdowns [191]. The adherence of the participants to the exercise interventions, as well as the effectiveness of these programs in improving several sleep parameters, highlight the emerging value of modern technology in promoting the well-being of cancer survivors.

The off-label use of over-the-counter dietary supplements and nutraceuticals is highly prevalent among patients with cancer, especially in hematologic malignancies [192]; nonetheless, there is currently no solid evidence to justify this practice [178,193]. Among the available supplements, vitamin D has gathered interest for its potential benefits in hematologic cancer patients, considering the high deficiency prevalence and association with adverse prognosis in this population [194]. Interestingly, a likely multi-level relationship also exists between vitamin D deficiency and increased body adiposity, although the value of exogenous supplementation in obesity outcomes remains controversial [193]. Regarding COVID-19 outcomes, data on the role of vitamin D deficiency as a risk factor for adverse outcomes are equivocal, while exogenous supplementation, although safe, has not provided concrete evidence towards a clear benefit [195].

## 10. Conclusions and Perspectives

It is likely that multiple factors related to obesity, hematologic malignancy or their mutual pathogenetic attributes predispose people to a more adverse SARS-CoV-2 infection prognosis, thus, forming a complex intercorrelated relationship. Moreover, the co-existence of both obesity and hematologic abnormalities in the frame of the SARS-CoV-2 infection may theoretically further adversely affect prognosis through independent risk factors representative of each condition. In any case, the societal and health care system-related adaptations during the course of the pandemic brought about a variety of obstacles pertinent to the diagnosis, timely active treatment, and chronic management of patients with hematologic cancers, given the unique inherent and therapeutical features of this morbidity spectrum.

Unfortunately, it is likely that the pandemic may have already taken its toll on this vulnerable patient group; recent global-scale data indicate that they may have delayed or missed MM diagnoses in the early COVID-19 era compared with the previous year, in conjunction with a decreased survival among newly diagnosed cases (HR 0.61, 0.38–0.81 for diagnoses in 2020 compared with 2019) [196].

The field of hemato-oncology has already overcome the initial blow to a great degree and adjusted to the new reality brought forward by COVID-19. Nonetheless, there is a need for continued appraisal and update of numerous aspects of hematologic malignancy management. Moreover, the contribution of obesity and its severity as an additional risk factor for severe COVID-19 among those with hematologic malignancies has not yet been investigated. Additionally, other clinical, laboratory and treatment-related factors predisposing to severe disease should be identified to effectively stratify patients in susceptibility risk groups. Optimal timing of hemato-oncologic therapy initiation, post-treatment follow-up and timely therapeutical planning, including allogeneic HSCT in case of infections that emerge during active medical therapy, also need to be determined. Likewise, more evidence is needed to establish a potential superior efficacy of specific vaccine types among patients with hematologic cancer with or without obesity, as well as an effective vaccination schedule in the long term. Lastly, more data are needed regarding the optimal medical management of SARS-CoV-2 infections in this patient group, under the consideration of respective diagnoses and treatments which may individually affect the natural history of COVID-19.

Hopefully, all the above-mentioned parameters will help establish uniform international recommendations for hemato-oncologic management during and after the COVID-19 era. These could, in turn, be utilized in shaping national and regional guidelines considering the different aspects and unique challenges of this field.

## Figures and Tables

**Figure 1 cancers-14-02494-f001:**
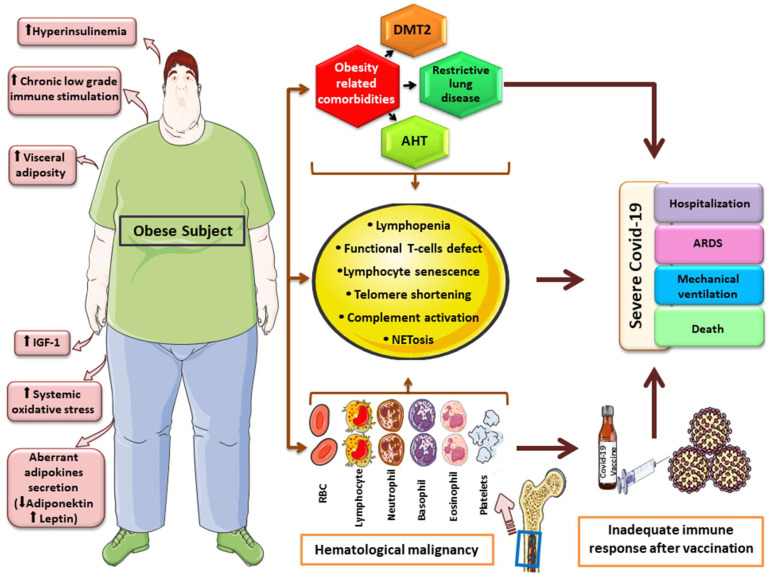
Graphical overview of factors and mechanisms linking obesity, hematologic malignancy and severe COVID-19 course. Obesity has been linked to carcinogenesis through aberrant cytokine and adipokine production, hyperinsulinemia, increase growth-factor levels (IGF-1), elevated oxidative-stress and chronic low-grade inflammation. Obesity-related comorbidities, along with features accompanying hematologic malignancies, such as quantitative and qualitative immune system defects and telomere shortening, may lead to an inadequate immune response after COVID-19 vaccination and severe clinical outcomes. * This image was derived from the free medical site http://smart.servier.com/ (accessed on 31 March 2022) by Servier, licensed under a Creative Commons Attribution 3.0 Unported License.

**Table 1 cancers-14-02494-t001:** Overview of obesity-related hematologic neoplasms.

Hematologic Malignancies	Study (Design, Reference)	Collective Main Findings
**Hodgkin Disease**	Meta-analysis [10] Population-based cohort study [11]	HR 1.41 (95% CI, 1.14–1.75) for BMI > 30 kg/m^2^ J-shaped association with BMI with ↑ incidence above 24.2 kg/m^2^↑ 10% HR per 5 kg/m^2^ increase
**Acute Leukemias** Lymphocytic Myeloid	Meta-analysis [12] Retrospective cohort study [13]	RR 1.65 (1.16–2.35) for BMI > 30 kg/m^2^ RR 1.52 (1.19–1.95) for BMI > 30 kg/m^2^
**Plasma cell disorders** MGUS Waldenström’s Makroglobulinemia Multiple Myeloma	Systematic Review [14] Population-based cohort study [15] Population-based cohort study [16] Prospective cohort study Prospective cohort study [17] Prospective cohort [18] Systematic review [19]	BMI associated MGUS development and progression to MM BMI > 30 kg/m^2^ associated with MGUS risk Obesity associated with MGUS progression to MM Viseral fat associated with LC-MGUS HR 1.41 (1.01–1.95) per 4 kg/m^2^ increase in BMI HR 1.41 (1.01–1.95) per 4 kg/m^2^ increase in BMI RR for MM mortality 1.71 ♂/1.44 ♀ for BMI > 35 kg/m^2^ vs. <25 kg/m^2^
**Non-Hodgkin’s Lymphoma/CLL**	Prospective cohort study [18] Meta-analysis [10] Population-based cohort study [20]	RR for NHL mortality 1.49 ♂/1.95 ♀ for BMI > 35 kg/m^2^ vs. <25 kg/m^2^ ↑ NHL incidence 1.07 (1.04–1.10)/1.14 (1.04–1.26) mortality per 5 kg/m^2^ increase in BMI, driven chiefly by ↑ risk of DLBCL CLL RR 1.25 (1.11–1.41) for BMI > 30 kg/m^2^
**Myelodysplastic syndromes**	Population-based cohort study [21] Retrospective cohort [22]	RR 2.18 (1.51–3.17) for BMI > 30 kg/m^2^ vs. <25 kg/m^2^ Higher BMI associated with lower overall survival, particularly in lower-risk MDS
**Chronic myeloproliferative disorders** Polycythemia Vera Chronic Myelogenous Leukemia Essential Thrombocythemia	Retrospective cohort study [23] Case–control study [24] Case–control study [25] Prospective cohort study [26]	BMI > 95th percentile among adolescents: aHR for PV 1.81 (1.13–2.92). Independent dose–response effect between adiposity and risk of CML OR for ET among obese 2.59 (1.02–6.58) RR for ET: 1.52 (0.97, 2.38) for BMI > 29.3 vs. <23.4 kg/m^2^

Abbreviations: ALL: acute lymphocytic leukemia; AML: acute myeloid leukemia; BMI: body mass index; CLL: chronic lymphocytic leukemia; CML: chronic myelogenous leukemia; DLBCL: diffuse large B-cell lymphoma; ET; essential thrombocytopenia; LC: light chain; NHL: non-Hodgkin lymphoma; HM: hematologic malignancy; aHR; (adjusted) hazard ratio; MGUS: monoclonal gammopathy of undetermined significance; MDS: myelodysplastic syndrome; MM: multiple myeloma; OR: odds ratio; PV: polycythemia vera; RR: relative risk.

**Table 2 cancers-14-02494-t002:** Overview of key studies addressing the impact of hematologic malignancies on COVID-19 outcomes.

Data Source, Reference	Population of Interest	Main Outcomes
Hematologic cancer registry of India [63]	565 reports of patients of all ages from tertiary Indian centers with HM and laboratory-confirmed COVID-19 between 21 March 2020–20 March 2021	↑ mortality (aHR 2.85, 1.58–5.13) and severe disease (aOR 2.73, 1.45–5.12 for AML vs. ALL) No differences between AML and other hematologic diagnoses ↑ mortality among those not in remission (aOR 1.85, 1.18–2.89) No effects of corticosteroid treatment or exposure to monoclonal antibodies
European Hematology Association Survey [64]	3,801 patients with HM and laboratory-confirmed COVID-19 from 132 hematology centers across Europe between March 2020–December 2020	Highest death rates in AML (40%) and MDS (42.3%) Active malignancy associated with ↑mortality (aHR 1.86, 1.62–2.14) Among different HL diagnoses, only AML independently associated with ↑mortality (aHR 2.046, 1.18–3.56 vs. NHL)
Nationwide retrospective study in Israel [65]	313 patients with HM and COVID-19 from 16 medical centers	Age > 70 years, arterial hypertension, active treatment associated with adverse outcomes Remdesivir treatment linked to ↓ mortality no effects of other treatment modalities (corticosteroids, enoxaparin, convalescent plasma)
Data from population-based registry in Madrid, Spain [66]	833 patients with HM and COVID-19 from 27 medical centers between 28 February 2020 and 25 May 2020	Overall, 62% severe/critical disease, 33% mortality (highest among AML and MDS patients, 40% and 42.3%, respectively) ↑ risk of death > 60 years, no effect of gender ↑ mortality for AML (aHR 2.22, 1.31–3.74 vs. NHL), Monoclonal antibody treatment and conventional chemotherapy (aHRs vs. nontreatment (aHRs 2.02, 1.14–3.60 and 1.50, 0.99–2.29 vs. no treatment, respectively) ↓ mortality for Ph-negative myeloproliferative disorders and treatment with hypomethylating agents (aHRs 0.33, 0.14–0.81 vs. NHL and 0.47, 0.23–0.94 vs. no treatment, respectively)
Case–control study from 2 Hospital in Wuhan province, China [4]	13 cases among 128 hospitalized patients with HM and 16 HCWs with COVID-19	↑ mortality for those with HM vs. controls (62% vs. 0, *p* = 0.002)
Meta-analysis of 34 studies in adult and 5 in pediatric populations [67]	3377 patients with HM from 39 studies in total	No effects of recent systemic overall antineoplastic or cytotoxic therapy (RRs 1.17, 0.83–1.64 and 1.29, 0.78–2.15 vs. no treatment, respectively) on COVID-19 mortality
Case control study from a nationwide database of patient electronic health records in the US [68]	73 million patients, 517.580 with 8 types of HMs, 420 with SARS-CoV-2 infection up to 1 September 2020	Significantly ↑ SARS-CoV-2 acquisition rates for HM vs. controls (overall aOR 11.9, 11.3–12.5 for diagnosis < 1 year, 2.3, 2.2–2.4 for prior diagnosis), highest among ALL, ET, MM, AML and lowest for PV ↑ Higher hospitalization and death rates for HM vs. non-HM
Prospective cohort study among patients enrolled UK Coronavirus Cancer Monitoring project [69]	227 patients with HM (Leukemia, Lymphoma, MM, others) among 1044 with active cancer and documented SARS-CoV-2 infection between 18 March 2020–8 May 2020	↑ risk for adverse outcomes for HM vs. solid tumor patients (aORs for high flow oxygen therapy 1.82, 1.11–2.94, NIV 2.10, 1.14–3.76, ICU 2.73, 1.43–5.11, severe/critical disease 1.57, 1.15–2.15) ↑ in-hospital mortality for HM patients who recently received chemotherapy (1.57, 1.15–2.15 vs. no recent chemotherapy)

Abbreviations: ALL: acute lymphocytic leukemia; AML: acute myeloid leukemia; ET; essential thrombocytopenia; NHL: non-Hodgkin lymphoma; HM: hematologic malignancy; HR; hazard ratio; MDS: myelodysplastic syndrome; MM: multiple myeloma; NIV: non-invasive ventilation; OR: odds ratio; PV: polycythemia vera; RR: relative risk.

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
