# Peer review of "Management of Hematologic Malignancies in the Era of COVID-19 Pandemic: Pathogenetic Mechanisms, Impact of Obesity, Perspectives, and Challenges"

_cancers, 2022, doi:10.3390/cancers14102494_

Round 1

Reviewer 1 Report

Dimitrios Tsilingiris et al, in their paper entitled "Management of hematologic malignancies in the era of COVID-19 pan-demic: pathogenetic mechanisms, impact of obesity, perspectives, and challenges" an interesting report . In my opinion the section 10 shortend 

Author Response

The authors would like to thank you for reviewing our manuscript. In line with your comment, the section 10 has been shortened by more than 20% in the revised version.

Reviewer 2 Report

Authors submitted an interesting narrative review article about two major factors, hematologic malignancies and obesity that play a pivotal role in leading to a more severe SARS-CoV-2 infection. They discussed the epidemiologic and pathogenic characteristics that both risk factors share. They also discussed the obstacles and ways of management in relevant to patients with obese and hematologic neoplasms. The introduction was written great and succinct. I have a concern about all abbreviations (HR, RR, BMI, etc.) you have in table 1 where they should have been written in full name, probably below the table. Additionally, it seems that the table is very busy and confused, so I believe that authors’ name in column 2 can be removed and you include only the study’s type and the reference’s number. In line 98, although it is well-known, I would prefer to add a justification for this statement “Despite its simplicity, BMI is an imperfect tool for the assessment of the qualitative features of adiposity”. In table 2, same comment regarding author name as table 1 should be applied. In line 166, you should start further from the margin. Regarding line 173-176, you have mentioned that “The risk of death in adults was found to be higher than in pediatric hematologic patients (34% vs. 4%) and among those, higher for those aged > 60 (Relative Risk-RR 1.8) as well as lower among white compared to non-white patients (RR 2.2), but it would be very interesting if you add a reason beyond that. Also, adding studies that scientifically propose why African Americans have higher rates of adverse effects compared to other ethnicities should be useful to understand the whole story. In line 231-2, Please delete this statement “for a detailed overview of the pathophysiology of COVID-19 refer to the review by Osuchowski et al”. In line 240, it would be better to discuss more about the association between lymphopenia and adverse SARS-CoV-2 outcomes. In line 300, it is suggested to change “Bruton kinase” to Bruton's tyrosine kinase (BTK). In line 301, it would be better to specify whose recipient of allogenic HSC have showed an acceptable antibody response after the vaccination; it should be interesting if you are able to add a previous study about autologous HSCT as well. I agree that limiting blood donation during the COVID-19 pandemic causes an extreme shortages of blood supply, so authors should suggest different ways to improve the current guidelines to be used down the road. In line 386, it would be great if authors can give examples of available drugs. In line 397, a typo mistake was found (“form” should be “from”). In line 431, RCT was mentioned, but It’d be better to give the full name. In lines 462 & 471, same comment for ASCT and HSCT. Regarding the conclusion, Tsilingiris et al. end the story with very interesting recommendations that might be helpful in the future. 

Author Response

Authors submitted an interesting narrative review article about two major factors, hematologic malignancies and obesity that play a pivotal role in leading to a more severe SARS-CoV-2 infection. They discussed the epidemiologic and pathogenic characteristics that both risk factors share. They also discussed the obstacles and ways of management in relevant to patients with obese and hematologic neoplasms. The introduction was written great and succinct.

Response: The authors would like to thank you for the thorough review of our manuscript.

I have a concern about all abbreviations (HR, RR, BMI, etc.) you have in table 1 where they should have been written in full name, probably below the table.

Response: Thank you for pointing out this omission. We have added an explanation of the used abbreviations as a table footnote. REVISED

Additionally, it seems that the table is very busy and confused, so I believe that authors’ name in column 2 can be removed and you include only the study’s type and the reference’s number.

Response: We have revised the table along the lines you suggested. REVISED

In line 98, although it is well-known, I would prefer to add a justification for this statement “Despite its simplicity, BMI is an imperfect tool for the assessment of the qualitative features of adiposity”.

Response: The section has been complemented as per your suggestion. REVISED

In table 2, same comment regarding author name as table 1 should be applied.

Response: We have revised the table along the lines you suggested. REVISED

In line 166, you should start further from the margin.

Response: We have made this correction. REVISED

Regarding line 173-176, you have mentioned that “The risk of death in adults was found to be higher than in pediatric hematologic patients (34% vs. 4%) and among those, higher for those aged > 60 (Relative Risk-RR 1.8) as well as lower among white compared to non-white patients (RR 2.2), but it would be very interesting if you add a reason beyond that.

Response: We have expanded on possible explanations for the presented findings. REVISED

Also, adding studies that scientifically propose why African Americans have higher rates of adverse effects compared to other ethnicities should be useful to understand the whole story.

Response: We have added a passage referencing studies pertinent to this issue in the revised version. REVISED

In line 231-2, Please delete this statement “for a detailed overview of the pathophysiology of COVID-19 refer to the review by Osuchowski et al”.

Response: The comment has been deleted. REVISED

In line 240, it would be better to discuss more about the association between lymphopenia and adverse SARS-CoV-2 outcomes.

Response: We have further expanded on the issue of the association between lymphopenia and COVID-19 outcomes in the current version. REVISED

In line 300, it is suggested to change “Bruton kinase” to Bruton's tyrosine kinase (BTK).

Response: The suggested change has been made. REVISED

In line 301, it would be better to specify whose recipient of allogenic HSC have showed an acceptable antibody response after the vaccination; it should be interesting if you are able to add a previous study about autologous HSCT as well.

Response: We have revised the relevant section along the lines you suggested. REVISED

I agree that limiting blood donation during the COVID-19 pandemic causes an extreme shortages of blood supply, so authors should suggest different ways to improve the current guidelines to be used down the road.

Response: We have added a paragraph with suggested strategies to counter blood supply shortages which may be considered in the future. REVISED

In line 386, it would be great if authors can give examples of available drugs. REVISED

Response: A concise report of indicated drugs is listed further in the same paragraph. We have revised the listed drugs based on the updated data of the cited “WHO living guideline” as of May 2022.  REVISED

In line 397, a typo mistake was found (“form” should be “from”).

Response: The typo has been corrected. REVISED

In line 431, RCT was mentioned, but It’d be better to give the full name.

Response: The referenced trial is mentioned with its unabbreviated title in the revised version.   REVISED

In lines 462 & 471, same comment for ASCT and HSCT.

Response: The term HSCT has been explained in line 336 of the revised version. We have thereafter used the terms “allogenous HSCT” and “autologous HSCT” to avoid further confusion with the abbreviation.   REVISED

Regarding the conclusion, Tsilingiris et al. end the story with very interesting recommendations that might be helpful in the future.

Response: Thank you very much for your kind comment.

Round 2

Reviewer 2 Report

The manuscript has been much improved over the first round. However, some minor comments are written below:

1- It would be suggested to remove the reference column in table 2 and add all reference numbers to the data source's columns.

2- Line 175 (previously 166) - You started further too much from the margin.

3- Lines 188-189 - This sentence "There appears to be a continuous and linear relationship between age and the risk of hospitalization of death due to SARS-CoV-2 infection, rather than a critical age cut-off pre-disposing to severe disease" need to be rephrased or corrected. 

4- Lines 464-469 - It would be better if you can add a reference, so the reader can know by who these drugs are recommended.  

Author Response

The manuscript has been much improved over the first round. However, some minor comments are written below:

Response: The authors would like to thank you once more for taking the time and effort to review our work.

1- It would be suggested to remove the reference column in table 2 and add all reference numbers to the data source's columns.

Response: The suggested change has been made. REVISED

2- Line 175 (previously 166) - You started further too much from the margin.

Response: The suggested change has been made. REVISED

3- Lines 188-189 - This sentence "There appears to be a continuous and linear relationship between age and the risk of hospitalization of death due to SARS-CoV-2 infection, rather than a critical age cut-off pre-disposing to severe disease" need to be rephrased or corrected.

Response: The sentence has been rephrased in order to communicate a clearer message. REVISED

4- Lines 464-469 - It would be better if you can add a reference, so the reader can know by who these drugs are recommended. 

Response: The presented indications on COVID-19 therapy are recommendations of the World Health Organization,  through its “living guideline on drugs for COVID-19”. This is referenced twice in the paragraph. In the revised version, we have additionally added the corresponding reference in the end of the sentence so that this can be clearly communicated. REVISED